# Correlation between Laboratory-Accelerated Corrosion and Field Exposure Test for High-Strength Stainless Steels

**DOI:** 10.3390/ma15249075

**Published:** 2022-12-19

**Authors:** Jinchao Jiao, Yong Lian, Zhao Liu, He Guo, Jin Zhang, Yan Su, Junpeng Teng, Yiming Jin, Jinyan Chen

**Affiliations:** 1Institute for Advanced Materials Technology, University of Science and Technology Beijing, Beijing 100083, China; 2Southwest Institute of Technology and Engineering, Chongqing 400039, China

**Keywords:** high-strength stainless steel, field exposure test, laboratory-accelerated corrosion, correlation, marine atmospheric environment

## Abstract

Equipment in a long-term marine atmosphere environment is prone to corrosion failure. Natural field exposure tests usually require a long time to obtain corrosion information. This study worked out a laboratory-accelerated corrosion test method that has a strong correlation with the natural environment test in Wanning, Hainan, and can be used as the basis for life assessment and the prediction of two high-strength stainless-steel materials. The mathematical model of corrosion weight loss of two high-strength stainless steels (3Cr13 and 00Cr12Ni10MoTi) was established by a field exposure test and a laboratory-accelerated corrosion test. Then, the correlation between the field exposure test and the laboratory-accelerated corrosion test was evaluated using qualitative and quantitative methods, and the acceleration ratio was calculated using the accelerated switching factor (ASF) method. The results show that: (1) The corrosion morphology of the two stainless steels after 15 days of laboratory-accelerated corrosion testing is similar to that obtained after two years of field exposure. (2) The value of gray correlation between the laboratory-accelerated corrosion test and the field exposure test is not less than 0.75. (3) The acceleration ratio of both stainless steels increases with the corrosion test time in the laboratory. The corrosion prediction models for the two stainless steels are T_3Cr13_ = 6.234 t^1.634^ and T_00Cr12Ni10MoTi_ = 55.693 t^1.322^, respectively.

## 1. Introduction

The exploration, collection and transportation of marine resources rely on offshore infrastructure and equipment. Due to the conditions of the marine environment, a large number of steel structural components of marine facilities are highly susceptible to corrosion [1,2,3]. Therefore, the development of high-strength, corrosion-resistant, low-cost materials, as well as reasonable structural design and material selection, has become one of the key technologies in the development of the infrastructure and equipment of marine facilities. High-load-bearing structural components require high-strength or even ultra-high-strength stainless-steel materials. Among the many types of stainless steels, martensitic stainless steel meets the requirements for high-strength structural components [4,5]. Currently, 13wt.% Cr-type martensitic stainless steel is the most used [6], but martensitic stainless steel contains a lot of carbides, so its corrosion resistance is not high [7,8,9,10]. Maraging stainless steel is a new series developed on the basis of maraging steel from the late 1960s. In addition to its high strength, high toughness, easy processing and forming, simple heat treatment and good welding performance of maraging steel [11,12,13], it also has excellent corrosion resistance [14,15].

It is essential to carry out research on the corrosion mechanism and regularity of stainless steel in the marine atmospheric environment. The traditional research method is the field exposure test in a natural environment [16], but due to the long period of the field exposure test, slow data recovery and the need for large manpower and material resources, the development of new corrosion-resistant materials has been severely limited [17]. Therefore, researchers expect to obtain atmospheric corrosion behavior and laws of materials in a relatively short period of time and predict the corrosion process of long-term outdoor exposure through short-term corrosion test results. At present, the most widely used is the laboratory-accelerated method, including the salt spray test [18,19,20], the wet–dry cycle immersion test [21,22,23,24,25,26], the cyclic corrosion test [27,28,29], etc. Whether the laboratory-accelerated test method is suitable for the studied material needs to be analyzed by correlating and establishing the connection between it and the external field exposure test [20,30,31,32]. The cyclic corrosion test includes three factors: salt spray, damp heat and dryness, which can better simulate the changes in the natural environment of the ocean atmosphere.

In this paper, martensitic stainless steel 3Cr13 and maraging stainless steel 00Cr12Ni10MoTi are the research objects. First, the corrosion weight-loss data of both stainless steels were obtained through the field exposure test and the laboratory-accelerated corrosion test. The laboratory-accelerated test parameters are based on the actual meteorological conditions in Wanning, Hainan, and the conversion method of the equivalent corrosion rate, which is the environmental load spectrum of Wanning, Hainan. The correlation between the laboratory-accelerated corrosion test and the field exposure test was then evaluated by qualitative and quantitative methods. Finally, the acceleration ratio of the laboratory-accelerated corrosion test to the two stainless-steel materials was calculated, and a corrosion-life prediction model based on the laboratory-accelerated corrosion test was established.

## 2. Materials and Methods

### 2.1. Materials

The experimental materials are martensitic stainless steel 3Cr13 (a commercial stainless steel offered by Daye Special Steel Co., Ltd., Huangshi, China) and maraging stainless steel 00Cr12Ni10MoTi, and their chemical components are shown in Table 1. Maraging stainless steel 00Cr12Ni10MoTi was prepared with low residual impurities in a 2-ton vacuum induction melting furnace, followed by vacuum arc remelting (Consarc, Inductotherm Group, Rancocas, NJ, USA). The heat treatment state and sample size of the two stainless steels are shown in Table 2. Before the start of the test, all samples were washed with absolute ethanol and dried with cold air, and the mass of the samples was weighed using an electronic balance (Mettler Toledo XPE1203S, Greifensee, Switzerland).

The specimens were polished to 2000# step by step using metallographic sandpaper and then polished with 1 μm grit diamond polish. 3Cr13 and 00Cr12Ni10MoTi were etched with 10% FeCl_3_ solution and Ralph reagent (100 mL H_2_O + 200 mL CH_3_OH + 100 mL HCl + 2 g CuCl_2_ + 7 g FeCl_2_ + 5 mL HNO_3_), respectively, and then metallographic photographs were taken using a metallographic microscope (Olympus Gx51, Tokyo, Japan). All chemical reagents used in the experiments were purchased from Sinopharm Chemical Reagent Co., Ltd., Shanghai, China.

### 2.2. Field Exposure Test

The field exposure test was conducted in accordance with the “GB/T 14165-2008 experimental standard”. The test station is located in Wanning City, Hainan Province, along the coast of Shangen Town, longitude 110°05′ E, latitude 18°58′ N, altitude 12.3 m. The annual average temperature and relative humidity are 23.9 °C and 85%, respectively. The annual sunshine duration is 2005 h, and the rainfall pH is 5.0. It is a typical high-temperature and high-humidity marine atmospheric environment. There were four groups of samples and corrosion times (0.5, 1, 1.5 and 2 years), with three samples in each group.

### 2.3. Laboratory-Accelerated Corrosion Test

The laboratory-accelerated corrosion test consists of three stages of salt spray, drying and damp heat. The detailed parameters of each stage are shown in Table 3. The experimental equipment model was the JX-FH-90 salt spray chamber (Jiaxi Experimental Instrument Co., Ltd., Shanghai, China). There were five groups of samples and corrosion times (3, 6, 9,12 and 15 days), with three samples in each group.

## 3. Results

### 3.1. Microstructure

Figure 1 shows the microstructures of two high-strength stainless steels. 3Cr13 martensitic stainless steel belongs to medium-carbon steel. After quenching, a mixed structure of lath (dislocation) martensite and flake (twin) martensite is obtained. During the tempering process, alloy carbides such as (Cr, Fe)_7_C_3_ are precipitated from the martensite structure to form a tempered troostite structure composed of an α phase and granular carbides that retain the characteristics of the original martensite (Figure 1a). 00Cr12Ni10MoTi maraging stainless steel is an ultra-low carbon stainless steel, and a lath martensite structure with high dislocation density can be obtained by solution treatment. After the aging treatment, the matrix structure still retains lath-like characteristics (Figure 1b). A small amount of massive austenite structure is found at the grain boundary, and the aging precipitates are small and cannot be identified in the metallographic structure.

Figure 2 shows the typical TEM microstructure and diffraction pattern of 00Cr12Ni10MoTi martensitic stainless steel after aging treatment at 510 °C. It can be seen from the bright-field image (Figure 2a) that a large number of needle-like precipitates are precipitated in the martensite matrix after aging treatment. The precipitation phase is a hexagonal crystal structure intermetallic compound η-Ni_3_Ti. Figure 2b is the dark-field image, with the arrow pointing to spot ${\left(22\bar{4}0\right)}_{\eta }$ in Figure 2c. The transmitted dark-field image clearly shows that the rod-shaped precipitates are dispersed in the lath martensite. The average diameter and length of the precipitates were about 3 nm and 12 nm, respectively. Figure 2d shows the complete diffraction pattern of the incident ${\left[011\right]}_{\alpha ’}$ crystal band. The diffraction spots are indexed and calibrated. It can be found that the $\left(22\bar{4}0\right)$ diffraction spots of the precipitates and the $\left(\bar{22}0\right)$ diffraction spots of the matrix are in a straight line, which indicates that η-Ni_3_Ti has the following orientation relationship with the martensite matrix:$${\left(011\right)}_{M}//{\left(0001\right)}_{\eta },\;{\left[\bar{22}2\right]}_{M}//{\left(22\bar{4}0\right)}_{\eta }$$

### 3.2. Corrosion Weight-Loss Kinetics

The corrosion weight-loss data of 3Cr13 and 00Cr12Ni10MoTi stainless steels in the field-exposure corrosion test and laboratory-artificial accelerated test are shown in Table 4 and Table 5, respectively. It can be seen that the corrosion weight loss of 00Cr12Ni10MoTi stainless steel is much smaller than that of 3Cr13 stainless steel in both test environments. As shown in Table 4, the corrosion weight loss of 3Cr13 martensitic stainless steel under the same exposure time is two orders of magnitude higher than that of 00Cr12Ni10MoTi maraging stainless steel, about 350 times.

The power function relationship model was used to fit the corrosion weight loss of the two materials in the natural environment test and the corrosion weight loss under the artificial acceleration test with time. A large number of studies have proved that the atmospheric corrosion kinetic equation of the power function model can reliably predict long-term corrosion. The empirical equation of the power function is as follows [33,34,35,36]:(1)$$D={\mathrm{At}}^{n}$$
where D is the corrosion weight loss (g/m^2^), t is the corrosion time (h), and A and n are constants. The value of n can reflect the characteristics of corrosion kinetics, that is, n>1 represents the corrosion acceleration process, n < 1 represents the corrosion deceleration process and n = 1 represents the uniform corrosion process [35,37].

Figure 3 shows the corrosion weight loss of two stainless steels with time and their fitting curves. D1 and D2 (Figure 3a) represent the fitting curves of the external field exposure corrosion weight loss and laboratory-accelerated corrosion weight loss of 3Cr13 martensitic stainless steel, respectively. D3 and D4 (Figure 3b) represent the fitting curves of the external field exposure corrosion weight loss and laboratory-accelerated corrosion weight loss of 00Cr12Ni10MoTi maraging stainless steel, respectively. The corresponding fitting equations are shown in Table 6. It can be seen that, under the field-exposed condition, the corrosion rate of both stainless steels decreases with time because their n values are less than 1. It should be noted that the 3Cr13 stainless steel is nearly uniformly corroded. In laboratory-accelerated corrosion, since the n value is greater than one, the corrosion rate of both stainless steels increased with time. It is shown that the laboratory acceleration method has an obvious acceleration effect on the two materials, and the decelerated corrosion process of external field exposure corrosion is transformed into an accelerated corrosion process. The reason may be that the short-term salt spray dry–wet alternation process during the accelerated corrosion process in the laboratory is not conducive to the “dissolution–reprecipitation” process of the corrosion product, which reduces the density of the corrosion product film, thereby reducing the corrosion product film’s protective effects. It can be seen from both field exposure and laboratory acceleration that the corrosion resistance of 00Cr12Ni10MoTi maraging stainless steel is better than that of 3Cr13 martensitic stainless steel in the atmospheric corrosion environment of Wanning, Hainan.

### 3.3. Corrosion Morphology

Figure 4 shows the corrosion morphologies of two stainless steels, 3Cr13 and 00Cr12Ni10MoTi, after being exposed to corrosion for 0.5 and 2 years. It can be seen that the surface of 3Cr13 was completely covered by corrosion products after half a year of corrosion (Figure 4a), and there were network cracks in the corrosion products. After two years of corrosion (Figure 4b), there was no obvious cracking on the surface of the corrosion product. 00Cr12Ni10MoTi had only a small amount of corrosion products of different sizes on the surface after being corroded for half a year (Figure 4c), showing obvious pitting corrosion characteristics. After two years of corrosion (Figure 4d), pitting pits could be observed on the surface of the sample, and there was a small amount of corrosion products around the pits. Therefore, from the appearance of corrosion products, the corrosion resistance of maraging stainless steel 00Cr12Ni10MoTi is better than that of martensitic stainless steel 3Cr13.

### 3.4. Corrosion Products

Figure 5 shows the XPS spectra of Fe 2p3/2 and Cr 2p3/2 of the corrosion products of two stainless steels. Figure 5a,c show the peak fitting results of the XPS spectrum of Fe 2p3/2. The spectrum of Fe 2p3/2 can be divided into three peaks, which are Fe^2+^ oxides, Fe^3+^ oxides and Fe^3+^ hydrated oxides. The corresponding corrosion products are FeO (710.7 eV), Fe_2_O_3_ (712.3 eV) and FeOOH (713.6 eV) [38]. Compared with 3Cr13, the FeO and Fe_2_O_3_ contents of 00Cr12Ni10MoTi are increased. The anodic reactions are as follows:(2)$$\mathrm{Fe}\;\to {\;\mathrm{Fe}}^{2+}+2e^{-}$$
(3)$${\mathrm{Fe}}^{2+}+2{\mathrm{OH}}^{-}\to \;\mathrm{Fe}{\left(\mathrm{OH}\right)}_{2}$$
(4)$$\mathrm{Fe}{\left(\mathrm{OH}\right)}_{2}\;\to \mathrm{FeO}+{\;H}_{2}O$$
(5)$$4\mathrm{Fe}{\left(\mathrm{OH}\right)}_{2}+{\;O}_{2}\;\to \;4\mathrm{FeOOH}+2H_{2}O$$
(6)$$2\mathrm{FeOOH}\to {\;\mathrm{Fe}}_{2}O_{3}+H_{2}O$$

Figure 5b,d show the peak fitting results of the XPS spectrum of Cr 2p3/2. Cr_2_O_3_ (575.2 eV), Cr(OH)_3_ (577.0 eV) and CrO_3_ (578.6 eV) are the main forms of Cr in corrosion products [39,40,41]. Compared with 3Cr13, the contents of Cr_2_O_3_ and Cr(OH)_3_ in 00Cr12Ni10MoTi increased significantly. Chromium plays an important role in the corrosion resistance of stainless steel, and chromium oxides are the main components of the passive film. The strength of Cr_2_O_3_ in 00Cr12Ni10MoTi is higher than that in 3Cr13, which improves the corrosion resistance. The formation of Cr(OH)_3_ and Cr_2_O_3_ is as follows [41]:(7)$$\mathrm{Cr}\;\to {\;\mathrm{Cr}}^{3+}+3e^{-}$$
(8)$${\mathrm{Cr}}^{3+}+3H_{2}O\;\to \;\mathrm{Cr}{\left(\mathrm{OH}\right)}_{3}+3H^{+}$$
(9)$$\mathrm{Cr}{\left(\mathrm{OH}\right)}_{3}\;\to {\;\mathrm{Cr}}_{2}O_{3}+2H_{2}O$$

Pitting corrosion is the most common corrosion type in high-strength stainless steel, and always occurs in the weak areas of passivation film, such as inclusions and carbide/intermetallic compound interfaces [42]. At the same time, the carbide precipitation in martensitic stainless steel causes local chromium depletion, which easily leads to intergranular corrosion [43]. Therefore, the difference in the corrosion resistance of the two stainless steels is related to their microstructure. During the tempering process of 3Cr13 martensitic stainless steel, alloy carbides such as (Cr, Fe)_7_C_3_ precipitate from the martensitic structure, resulting in the existence of a Cr-depleted zone in the carbide boundary region [44]. Martensitic stainless steels passivate spontaneously by forming an oxide layer consisting mainly of Fe and Cr, so the precipitation of Cr-rich carbides reduces the passivation film and pitting resistance [6,7,10,45,46,47]. In the presence of aggressive anions (especially chloride ions), the breakdown of the passivation film causes the exposed metal to dissolve and thus form corrosion pits [6,7]. A corrosion cell is formed between the corrosion pit (anode) and the passivation film (cathode) surrounding the pit. After the aging treatment, the 00Cr12Ni10MoTi maraging stainless steel has a lath martensite structure with a small amount of reversed austenite. There are a large number of nano-scale intermetallic compounds of η-Ni_3_Ti dispersed in the lath martensite, which will not form chromium-rich carbides. The formation of reversed austenite reduces the consumption of Cr around the carbide and the appearance of the Cr-depleted zone, which, in turn, enhances the passive film stability and pitting resistance [48,49]. In addition, the high content of Ni and Mo in the reversed austenite is beneficial to improve the pitting resistance in the austenite zone [50]. It has been reported that Ni can positively shift the pitting potential of stainless steel and contribute to the formation of Cr_2_O_3_ in the passivation film [51,52]. Therefore, 00Cr12Ni10MoTi martensitic stainless steel has better corrosion resistance in the specified environment.

## 4. Correlation Analysis

### 4.1. Qualitative Analysis

Qualitative analysis was performed using the method of macroscopic morphological comparison of corrosion specimens. The specimens exposed to corrosion in the external field for 2 years and the specimens exposed to accelerated corrosion in the laboratory for 15 days were selected for appearance comparison. As shown in Figure 6, when comparing the macroscopic corrosion morphology of the specimens under the two experimental environments, it can be seen that the surface of 3Cr13 martensitic stainless steel is covered by corrosion products, while the surface of 00Cr12Ni10MoTi martensitic stainless steel presents uniformly distributed pitting corrosion products. As the laboratory-accelerated corrosion accelerated the corrosion process, and due to the high moisture in the test chamber, a large number of liquefied solutions form the corrosion products and flow more obviously. Therefore, from the perspective of corrosion, the macroscopic morphology of laboratory-accelerated corrosion and external-exposure corrosion has a good consistency. The macroscopic corrosion morphology also indicates that the corrosion resistance of 00Cr12Ni10MoTi martensitic stainless steel is better than that of 3Cr13 martensitic stainless steel.

### 4.2. Quantitative Analysis

Gray correlation analysis is used to measure the degree of association between two factors or two systems [32,53]. In this work, it was used to analyze the correlation between the laboratory-accelerated and field exposure tests of two stainless steels.

The weight-loss data from the field exposure test were used as a reference series to calculate the correlation between the laboratory-accelerated corrosion test and the field exposure test [54]. The calculation formula is below:(10)$$\xi_{i}\left(k\right)=\frac{\underset{i}{\min}\;\underset{k}{\min}|x_{0}\left(k\right)-x_{i}\left(k\right)|+\rho \;\underset{i}{\max}\;\underset{k}{\max}|x_{0}\left(k\right)-x_{i}\left(k\right)|}{|x_{0}\left(k\right)-x_{i}\left(k\right)|+\rho \;\underset{i}{\max}\;\underset{k}{\max}|x_{0}\left(k\right)-x_{i}\left(k\right)|}$$
where ρ is the resolution factor, which is generally set to 0.5. The calculation is performed using the homogenization method. The calculation results are shown in Table 7.

In general, when the gray correlation is greater than 0.6, the sequence is considered to be closely related to the reference sequence [32]. From Table 7, the gray correlation values between the laboratory-accelerated experiments and the external exposure tests for 3Cr13 martensitic stainless steel and 00Cr12Ni10MoTi martensitic stainless steel are 0.82 and 0.75, respectively, indicating a good correlation.

In summary, the study by qualitative and quantitative analysis showed a good correlation between the accelerated-laboratory test and the external field exposure test.

### 4.3. Acceleration Ratio Analysis

The accelerated conversion factor (ASF) method was used to calculate the acceleration ratio. The corrosion weight-loss data obtained in the laboratory-accelerated corrosion test were substituted into the kinetic equation of the field exposure test to calculate the time required for the field exposure test. Thereby, the relationship between the laboratory-accelerated corrosion time and the field exposure time was established. The calculation results are shown in Table 8.

The function between the accelerated corrosion time and the field exposure time for the two stainless steels was obtained by regression analysis as:(11)$$T_{3\mathrm{Cr}13}=6.234{\;t}_{\;}^{1.634}\;R^{2}=0.984$$
(12)$$T_{00\mathrm{Cr}12\mathrm{Ni}10\mathrm{MoTi}}=55.693{\;t}_{\;}^{1.322}\;R^{2}=0.985$$
where t is the laboratory acceleration time (days) and T is the field exposure time corresponding to t (days).

From this, the acceleration ratio of the two materials is deduced:(13)$${\mathrm{ASF}}_{3\mathrm{Cr}13}=T_{3\mathrm{Cr}13}/t=6.234{\;t}_{\;}^{0.634}$$
(14)$${\mathrm{ASF}}_{00\mathrm{Cr}12\mathrm{Ni}10\mathrm{MoTi}}=T_{00\mathrm{Cr}12\mathrm{Ni}10\mathrm{MoTi}}/t=55.693{\;t}_{\;}^{0.322}$$

Equations (2)–(5) are plotted in Figure 7. From Figure 7b, it can be seen that the acceleration ratios of both 00Cr12Ni10MoTi and 3Cr13 increase with the increase in accelerated corrosion time. During this accelerated corrosion test, the acceleration ratio of 3Cr13 ranged from 0 to 50, and that of 00Cr12Ni10MoTi ranged from 50 to 150.

## 5. Conclusions

(1)The corrosion resistance of 00Cr12Ni10MoTi martensitic stainless steel is greatly improved compared with that of 3Cr13 martensitic stainless steel, and the corrosion kinetics fitting curve is in accordance with the power function law.(2)Qualitative and quantitative analysis results show that the laboratory-accelerated corrosion test and the external field exposure test have a good correlation where the gray correlation coefficient is greater than 0.75. Therefore, the artificially accelerated corrosion method used in this study can be used as the basis for life assessment and the prediction of stainless-steel materials. The corrosion prediction models for the two stainless steels are $T_{3Cr13}=6.234\;t^{1.634}$ and $T_{00\mathrm{Cr}12\mathrm{Ni}10\mathrm{MoTi}}=55.693\;t^{1.322}$, respectively.(3)The passivation film and corrosion product film formed on the surface of 00Cr12Ni10MoTi martensitic aging stainless steel has a protective effect, while the passivation film and corrosion product film on the surface of 3Cr13 martensitic stainless steel has a lower protective effect.

## Figures and Tables

**Figure 1 materials-15-09075-f001:**
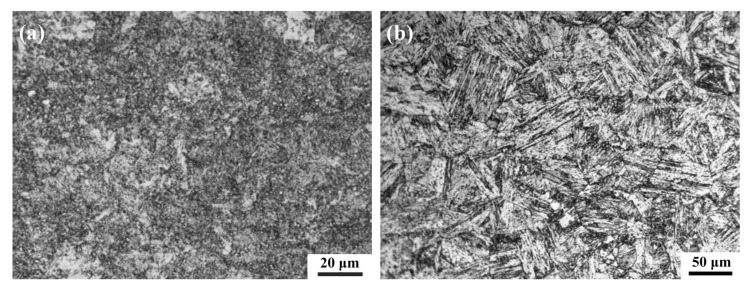
Microstructure of two materials: (**a**) 3Cr13 martensitic stainless steel tempered at 540 °C, (**b**) 00Cr12Ni10MoTi martensitic stainless steel aged at 510 °C.

**Figure 2 materials-15-09075-f002:**
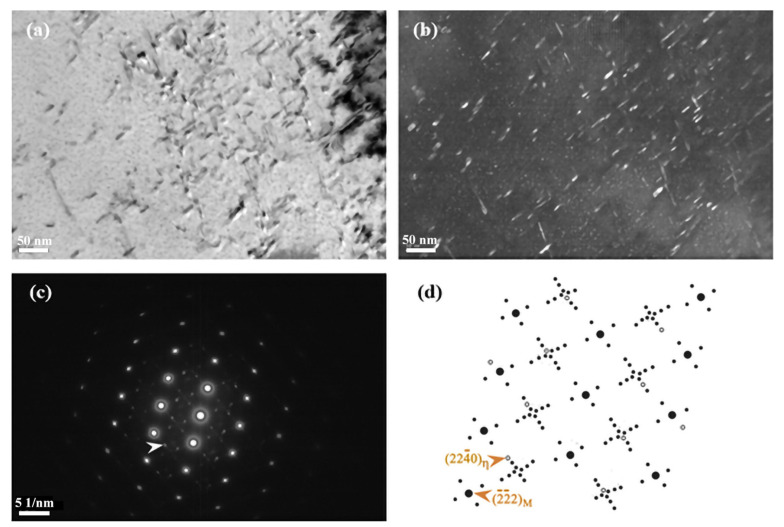
TEM photo of 00Cr12Ni10MoTi maraging stainless steel aged at 510 °C for 4 h: (**a**) bright-field image, (**b**) dark-field image taken from the diffraction spot indicated by an arrow in (**c**) showing precipitates, (**c**) selected area diffraction pattern with zone axis ${\left[011\right]}_{\alpha ’}$, (**d**) schematics of selected area diffraction pattern with zone axis ${\left[011\right]}_{\alpha ’}$.

**Figure 3 materials-15-09075-f003:**
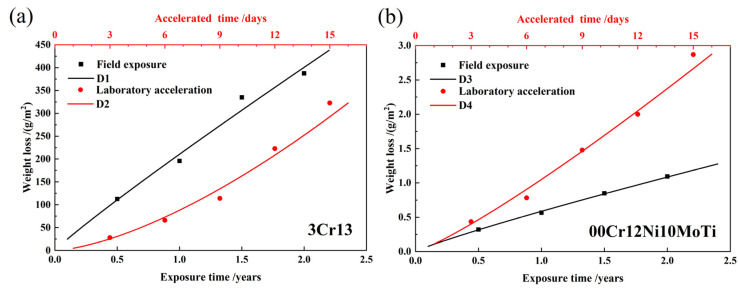
Field exposure and laboratory-accelerated corrosion weight loss of two materials with time and their fitting curves: (**a**) 3Cr13, (**b**) 00Cr12Ni10MoTi.

**Figure 4 materials-15-09075-f004:**
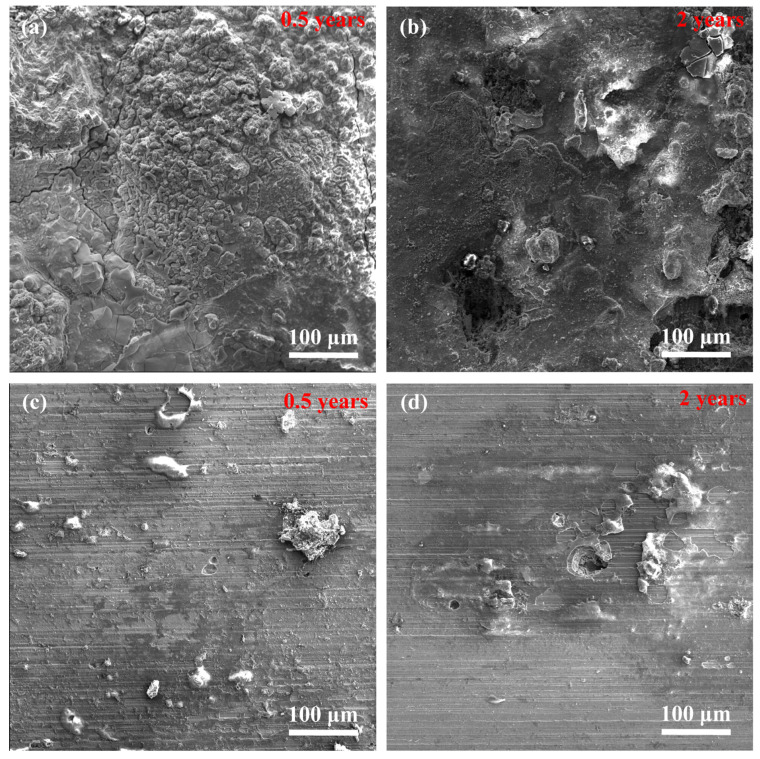
Corrosion microscopic morphology of two stainless steels after field exposure for different times: (**a**,**b**) 3Cr13, (**c**,**d**) 00Cr12Ni10MoTi.

**Figure 5 materials-15-09075-f005:**
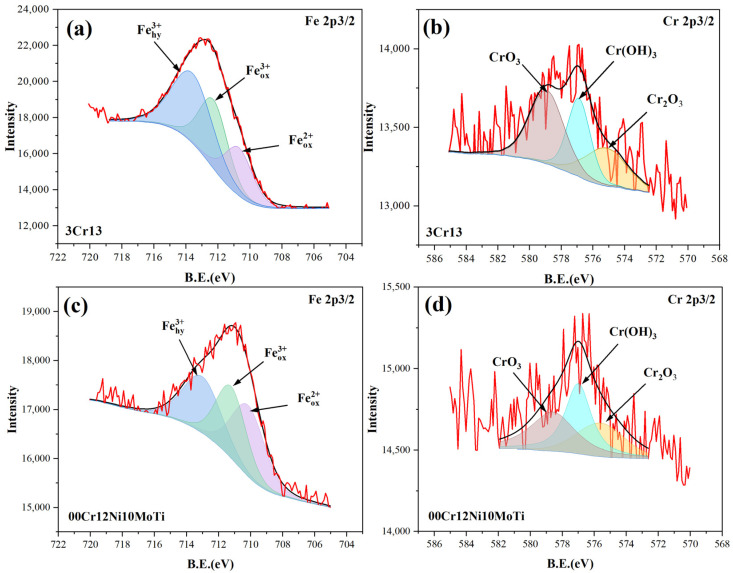
XPS spectra of corrosion products of two stainless steels corroded for two years: (**a**,**b**) 3Cr13, (**c**,**d**) 00Cr12Ni10MoTi.

**Figure 6 materials-15-09075-f006:**
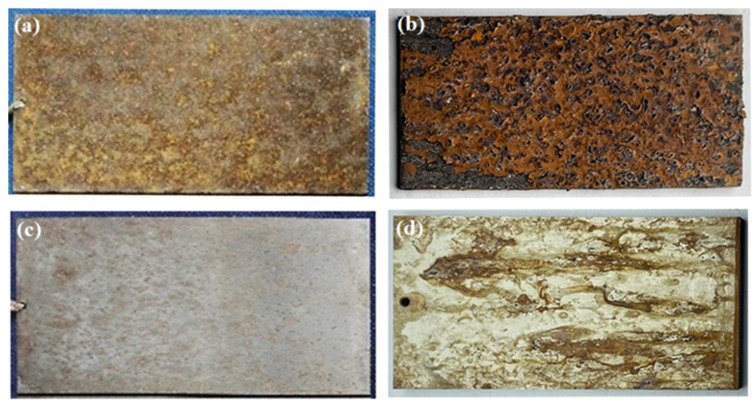
Comparison of corrosion morphology of two materials. Field exposure corrosion test for 2 years (**a**) and laboratory-accelerated corrosion test for 15 days (**b**) with 3Cr13; field exposure corrosion test for 2 years (**c**) and laboratory-accelerated corrosion test for 15 days (**d**) with 00Cr12Ni10MoTi.

**Figure 7 materials-15-09075-f007:**
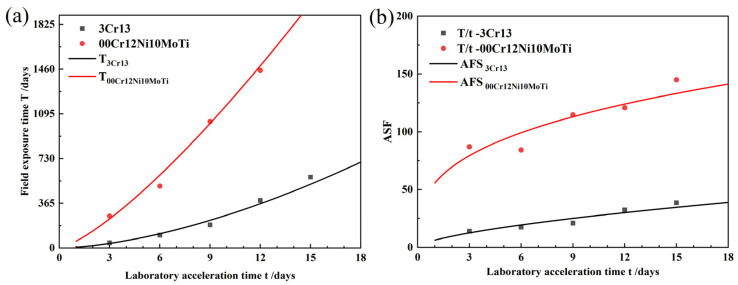
Acceleration ratio analysis: (**a**) relationship between laboratory-accelerated corrosion time and field exposure time, (**b**) acceleration ratio.

**Table 1 materials-15-09075-t001:** Chemical composition content of two high-strength stainless steels (wt.%).

Material	C	Cr	Mo	Ni	Ti	Mn	Si	P	S	Fe
3Cr13	0.25–0.34	12.0–14.0	/	≤0.6	/	≤0.6	≤0.6	≤0.030	≤0.030	Bal.
00Cr12Ni10MoTi	≤0.03	11.0–13.0	0.5–2.0	9.5–12.0	0.8–2.0	/	/	≤0.015	≤0.015	Bal.

**Table 2 materials-15-09075-t002:** Sample state and size.

Material	Heat Treatment	Sample Size
3Cr13	Quenching at 980 °C and tempering at 540 °C	100 mm × 50 mm × 3 mm
00Cr12Ni10MoTi	Solution at 980 °C and aging at 510 °C	

**Table 3 materials-15-09075-t003:** Laboratory-accelerated test conditions.

Step	Duration Time (h)	Temperature (°C)	Relative Humidity	Solution (wt.%)
Salt spray	16	40	/	5% NaCl + 0.1% Na_2_SO_4_ pH = 4
Dry	1	60	<30%	/
Wet	7	40	90%	/

**Table 4 materials-15-09075-t004:** Corrosion weight-loss data of two stainless steels in natural environment.

Exposure Time (Years)	Corrosion Weight Loss (g·m^−2^)	
	3Cr13	00Cr12Ni10MoTi
0.5	112.594	0.321
1.0	195.689	0.566
1.5	334.926	0.849
2.0	387.453	1.094

**Table 5 materials-15-09075-t005:** Corrosion weight-loss data of two stainless steels in laboratory artificially accelerated environment.

Accelerated Test Time (Days)	Corrosion Weight Loss (g·m^−2^)	
	3Cr13	00Cr12Ni10MoTi
3	28.087	0.435
6	66.000	0.783
9	114.000	1.478
12	222.870	2.000
15	322.783	2.870

**Table 6 materials-15-09075-t006:** Corrosion kinetic equations of two stainless steels.

Material	Experimental Conditions	Functional Model	Unit of t	R^2^
3Cr13	Field exposure	$D1=210.470{\;t}^{0.929}$	years	0.985
	Laboratory acceleration	$D2=4.780{\;t}^{1.518}$	days	0.984
00Cr12Ni10MoTi	Field exposure	$D3=0.586{\;t}^{0.889}$	years	0.998
	Laboratory acceleration	$D4=0.110{\;t}^{1.177}$	days	0.984

**Table 7 materials-15-09075-t007:** Grey correlation between laboratory-accelerated and field exposure test.

Material	Raw Weight Loss (g·m^−2^)		Homogenization		Correlation Degree ξ_i_	Mean
	Field Exposure	Laboratory-Accelerated	Field Exposure	Laboratory-Accelerated		
3Cr13	112.594	28.087	0.437	0.261	0.94	0.82
	195.689	66.000	0.759	0.613	1.00	
	334.962	114.000	1.300	1.058	0.82	
	387.453	222.870	1.504	2.069	0.51	
00Cr12Ni10MoTi	0.321	0.435	0.454	0.152	1.00	0.75
	0.566	0.783	0.800	0.273	0.76	
	0.849	1.478	1.200	0.515	0.66	
	1.094	2.000	1.546	0.697	0.57	

**Table 8 materials-15-09075-t008:** Relationship between laboratory-accelerated corrosion time and field exposure time for two stainless steels.

Material	Accelerated Time (Days)	Weight Loss (g·m^−2^)	Functional Model	Time of Field Exposure (Days)
3Cr13	3	28.087	$D1=210.470t^{0.929}$	42
	6	66.000		105
	9	114.000		189
	12	222.870		388
	15	322.783		578
00Cr12Ni10MoTi	3	0.435	$D3=0.586t^{0.890}$	261
	6	0.783		505
	9	1.478		1032
	12	2.000		1450
	15	2.86957		2175

## Data Availability

Not applicable.

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
