# Peer review of "Correlation between Laboratory-Accelerated Corrosion and Field Exposure Test for High-Strength Stainless Steels"

_materials, 2022, doi:10.3390/ma15249075_

Round 1

Reviewer 1 Report

The article is written at a very good level. The problem of corrosion behavior of two corrosion-resistant materials is solved here. The authors appropriately chose testing methods to create a mathematical model of the corrosion rate of the given materials for the given corrosion environment.

There are only a few ambiguities or possible additions in the text:

line 91 - here the corrosion time is described as 0.5 a, 1 a, 1.5 a and 2 a. It is not clear from the text what time unit the authors have in mind. and....does it mean hours, days, years? This must be specified in the text for better orientation and comprehensibility. The same applies to the corrosion time denoted as d.

Table 4 and Table 5 Again confusion in the units Exposure time and Accelerated test time?

line 215 - at the end of the sentence add....corrosion resistance in the specified environment.

line 289 - it would be better to use has and a lower protective effect.

Reviewer 2 Report

1. In paragraph 2, it is necessary to add manufacturing companies and countries for each material, chemical reagent, and equipment used.

2. The article lacks chemical and electrochemical reactions of the corrosion process.

3. Based on the corrosion rates, chemical, phase composition, and morphology, it is logical to add a corrosion mechanism or explain in more detail the effect of the dechromization process on the course of the corrosion process.

Round 2

Reviewer 2 Report

Good job